# COVID and Cancer: A Complete 3D Advanced Radiological CT-Based Analysis to Predict the Outcome

**DOI:** 10.3390/cancers15030651

**Published:** 2023-01-20

**Authors:** Syed Rahmanuddin, Asma Jamil, Ammar Chaudhry, Tyler Seto, Jordyn Brase, Pejman Motarjem, Marjaan Khan, Cristian Tomasetti, Umme Farwa, William Boswell, Haris Ali, Danielle Guidaben, Rafay Haseeb, Guibo Luo, Guido Marcucci, Steven T. Rosen, Wenli Cai

**Affiliations:** 1Department of Diagnostic Radiology, City of Hope Comprehensive Cancer Center National Medical Center, Duarte, CA 91010, USA; 2Department of Quality, Risk and Regulatory Management, City of Hope National Medical Center, Duarte, CA 91010, USA; 3Center for Cancer Prevention and Early Detection, City of Hope National Medical Center, Duarte, CA 91010, USA; 4Department of Hematology-Medical Oncology, City of Hope National Medical Center, Duarte, CA 91010, USA; 5Department of Radiology, Massachusetts General Hospital, Harvard University, Boston, MA 02114, USA

**Keywords:** cancer, COVID-19, CT scan, corona virus, treatment

## Abstract

**Simple Summary:**

Coronavirus-19 is caused by SARS-CoV-2 which is a fatal disease if untreated. The chance of mortality is higher in cancer patients because, in immunocompromised cancer patients, lung damage occurs more rapidly than in normal non-cancer patients which needs more precision based diagnostic criteria. Our study highlights the significance of CT-based radiomics findings to detect morphological damage in the lungs due to COVID-19 in cancer patients. This methodology will help clinicians to investigate the disease more precisely for more accurate precision-based diagnosis.

**Abstract:**

Background: Cancer patients infected with COVID-19 were shown in a multitude of studies to have poor outcomes on the basis of older age and weak immune systems from cancer as well as chemotherapy. In this study, the CT examinations of 22 confirmed COVID-19 cancer patients were analyzed. Methodology: A retrospective analysis was conducted on 28 cancer patients, of which 22 patients were COVID positive. The CT scan changes before and after treatment and the extent of structural damage to the lungs after COVID-19 infection was analyzed. Structural damage to a lung was indicated by a change in density measured in Hounsfield units (HUs) and by lung volume reduction. A 3D radiometric analysis was also performed and lung and lesion histograms were compared. Results: A total of 22 cancer patients were diagnosed with COVID-19 infection. A repeat CT scan were performed in 15 patients after they recovered from infection. Most of the study patients were diagnosed with leukemia. A secondary clinical analysis was performed to show the associations of COVID treatment on the study subjects, lab data, and outcome on mortality. It was found that post COVID there was a decrease of >50% in lung volume and a higher density in the form of HUs due to scar tissue formation post infection. Conclusion: It was concluded that COVID-19 infection may have further detrimental effects on the lungs of cancer patients, thereby, decreasing their lung volume and increasing their lung density due to scar formation.

## 1. Introduction

Severe acute respiratory syndrome coronavirus-2 (SARS-CoV-2) is a novel coronavirus that emerged from Wuhan, China, which can cause severe respiratory illness and a myriad of other complications now referred to as COVID-19. COVID-19 became a pandemic with more than three million confirmed cases and more than 220,000 deaths around the globe [1]. It can lead to severe symptoms ending in hospitalization, often due to acute respiratory distress syndrome (ARDS) [2,3,4].

In addition, cancer affects a severely vast proportion of the population around the world contributing to more than 18 million cases recorded every year. Individuals with cancer are more prone to infections because of their poor immune and health status due to the prevailing cancer and its anticancer treatment [5]. To date, cancer patients have been considered to be one of the most vulnerable groups in the current COVID-19 pandemic. Although cancer patients infected with SARS-CoV-2 have been shown, in a multitude of studies, to have poor outcomes on the basis of older age and weakened immune systems from cancer as well as chemotherapy [3,4,5,6,7,8,9,10,11,12], nevertheless, the clinical characteristics of SARS-CoV-2-infected cancer patients remain largely unknown [8]. 

In addition, cancer patients have increased contact with healthcare systems as they visit hospitals for monitoring, anticancer therapy, as well as supportive and preventive care [13].

Recently, a study was conducted among cancer patients to determine the severity of COVID-19 as compared with patients without cancer [8]; however, due to the limited sample size of this study, the results were not applicable to the overall population, and thereby, no insightful analyses were performed [14,15]. Another study concluded that COVID-19 infection in cancer patients posed increased risk for more deteriorating conditions and severe health outcomes [8]. Another study, conducted in China, revealed that as compared with non-cancer patients, COVID-19 cancer patients had less favorable outcomes and a higher risk of disease course with increased severity [6,7,8,9,10]. 

In another study, the authors concluded that, as per the results of epidemiological studies in which the majority of patients included in the COVID-19 cohort also had cancer, cancer patients were more likely to develop COVID-19 infection [14]. In contrast, another study concluded that COVID-19 was a very contagious infection to which almost every individual was susceptible, once they were exposed to an infection source [16].

Additionally, it has been concluded by researchers that, in this COVID-19 outbreak, the major risk for the morbidity and mortality of the infected individuals was access to medical care services in terms of access to a healthcare facility and provision of best possible treatment [17]. Moreover, increased severity and ultimate death from COVID-19 in cancer patients were principally driven by gender; age; need for ICU support; elevated levels of D-dimer, lactate dehydrogenase, and lactate; and comorbidities such as diabetes mellitus, liver disease, cardiovascular disease, chronic kidney disease, respiratory disease, and immunosuppression in multivariate analyses [11,18,19,20].

Furthermore, hematologic and lung cancer patients are more susceptible. In addition, patients in the metastatic stages of their cancer have a higher frequency of severity as compared with patients without cancer. Further, patients who have undergone cancer surgery have increased mortality rates and increased chances of devastating symptoms [9].

Among COVID-19 cancer patients, high all-cause mortality associated with both general and unique risk factors related to cancer have been reported. In this study, the CT scans of 13 confirmed COVID-19 cancer patients were analyzed to determine the CT scan changes before and after treatment and the extent of structural damage to the lungs after COVID-19 infection in terms of changes in density and lung volume reduction. The main purpose of this study is to see the COVID-related morphological changes in the lungs of cancer patients who already have immune deficiencies. Cancer patients have less immune response and receive multiple chemotherapies which places them in the category of high-risk individuals for COVID-related diseases. Our research aim is to analyze the CT-based precision imaging findings to accurately estimate the damage in the lungs which could be helpful in clinical diagnosis & management. 

## 2. Materials and Methods

### 2.1. Patients

This retrospective study was approved by the City of Hope National Medical Center Comprehensive Cancer Center Institutional Review Board. A total of 28 cancer patients were recruited for the study, including 18 males (65.5%) and 10 females (34.5%) with ages ranging from 18 to 83 years and an overall mean age of 55.6 ± 18.54 years. The data was collected from City of Hope patients during the period from January 2020 to June 2021. All adult patients who were identified with COVID-19 infection were included in this study; all others were excluded.

### 2.2. Study Procedure

A retrospective analysis was conducted on the 28 cancer patients. The CT scan changes before and after treatment and the extent of structural damage to the lungs after COVID-19 infection were analyzed. Structural damage to the lung was indicated by changes in density measured in Hounsfield units (HUs) and lung volume reduction. A 3D radiometric analysis of data was also performed, and density histograms were constructed. Further, secondary clinical analysis was also completed to show the association of COVID-19 treatment on the study subjects, lab data, and outcome on mortality.

### 2.3. Methods

The data was collected from City of Hope patients during the period of 01 January 2020 till 30 June 2021. Volumetric quantification of lung and pneumonia lesions was carried out using the deep-learning segmentation model of lung and lesions on the 3DQI platform (https://3dqi.mgh.harvard.edu, accessed on 30 June 2021) [18]. After automated segmentation of lung and lesions, an image analyst, who was blinded to clinical data, reviewed and confirmed the segmentation results. A 3D radiometric analysis of the data was performed on the 3DQI Radiometrics tool based on extracted image features and clinical data. 

### 2.4. Statistics and Data Analysis

Demographic, pathological, and clinical characteristics of the patients were summarized using frequencies and percentages for categorical variables; however, mean and standard deviation (mean ± SD) were obtained for continuous variables. The potential changes in the patients were analyzed using means, standard deviations, and kurtosis values. Statistical analyses were conducted using IBM SPSS, version 20 (Statistical Package for the Social Science, IBM Corp, Armonk, NY, USA). 

## 3. Results

### 3.1. Patients’ Characteristics

The patients’ demographic characteristics are summarized in Table 1. The patients’ ages ranged from 18 to 83 years and the overall mean age was 55.6 ± 18.54 years.

### 3.2. Metastatic Profile of Cancer Patients

A CT image analysis was performed to analyze structural damage to the lungs of the cancer patients. Structural damage to the lung was indicated by changes in density measured in Hounsfield units (HUs) and lung volume reduction (Table 2). 

### 3.3. Metastatic Profile of Cancer Patients Post COVID-19 Infection

A total of 22 cancer patients diagnosed with COVID-19 infection were included; most of the study patients were also diagnosed with leukemia. A repeat CT scan was performed in the patients after they recovered from infection. It was found that post COVID there was a decrease of >50% in lung volume and higher density in the form of HUs due to scar tissue formation post infection (Table 3).

Statistical analysis (Table 4) and 3D radiometric analysis of data were also performed, and density histograms were also constructed for the patients (Figure 1, Figure 2, Figure 3, Figure 4 and Figure 5). To analyze the changes in lung volume and lesion volumes between the pre- and post-treatment, we calculated the change in the percentage of lung volume, the change in the lesion volume, and the change in lesion percentage of each patient. We have reported the means and standard errors.

## 4. Discussion

Due to the rapid spread of the COVID-19 infection around the world, many countries have not been prepared to handle the large number of people affected due to a lack of awareness about the general pathophysiology of the COVID-19 infection and its impact on individuals’ lives, in addition to weak health systems and poor access of people to healthcare facilities [9]. Cancer patients are not only more likely to be exposed to COVID-19 due to their need to access the healthcare system, but they are also more susceptible to the disease itself. For instance, COVID-19 cancer patients experience more severe outcomes as compared with the general population both in terms of morbidity and mortality rates. In the general population, the mortality rate for the COVID-19 infection is only 2–3%; however, it increases up to three fold in COVID-19 cancer patients [7,21].

Factors associated with increased severity and ultimately death from COVID-19 in cancer patients include gender; age; need for ICU support; elevated levels of D-dimer, lactate dehydrogenase, and lactate; as well as comorbidities [11].

Previous studies have analyzed the severity and case fatality of cancer patients if co-infected with COVID-19 and have also found that male sex was the prominent risk factor for the infection [20,22]. Other researchers have also concluded that there is a higher incident and mortality rate of COVID-19 infections in males in general. Although the pathogenesis behind this is not established yet, several hypotheses have been proposed [23,24]. For instance, a hypothesis was established that the androgenic hormones have a potential role in the pathogenesis of COVID-19 infection in cancer patients, thereby, posing a higher risk in males for severe outcomes [25]. Another hypothesis proposed that the sex differences in the multistep immune-pathogenic pathway among both males and females leads to sex biases causing the male gender to be more prone to the COVID-19 infection [26,27]. Because none of these theories have been accepted or established in the medical literature, scientists are still searching for the most appropriate reason to explain why male sex appears to be a risk factor for COVID-19 cancer patients [28]. 

Similar to our study, a study conducted in Wuhan China among COVID-19-infected lung cancer patients found reduced lung volume due to the presence of tumor and also the features of pneumonia through CT scan images [8]. This study concluded that COVID-19-infected lung cancer patients had worse endurance and lung function at baseline and were prone to develop anoxia which progressed more rapidly due to COVID-19, thereby, creating an urgent need for the treatment of COVID-19-infected lung cancer patients [8]. Further, it was concluded that decreased lung function and severe infection in lung cancer patients contributed to the more severe and devastating outcomes in COVID-19 cancer patients [29,30]. Moreover, patients with lung cancer are more likely to deteriorate in a very short span of time when infected with COVID-19 infection due to [9] pulmonary edema, focal reactive hyperplasia of pneumocytes, proteinaceous exudate, hyaline membrane formation, and multinucleated giant cells with patchy inflammatory cellular infiltration.

In this retrospective analysis, the CT scans of 22 confirmed COVID-19 patients with leukemia were analyzed to determine the changes before and after treatment and the extent of structural damage to the lungs after COVID-19. Structural damage to the lung was indicated by changes in density measured in Hounsfield units (HUs) and lung volume reduction. Repeat CT scans were performed in 15 of the patients after they recovered from infection. It was found that, post COVID, there was a decrease of >50% in lung volume and a higher density in the form of HUs due to scar tissue formation post infection.

Similar to other studies of COVID-19, the current study also showed increased cases of males (65.5%) as compared with females (34.5%). Similar to previous findings [19,20], the current results also indicated that the increased severity of COVID-19 cancer patients was related to older age as most of our patients were above 50 years of age. This might be due to the fact that in older age, the host response is diminished and the immune system is compromised and suffering from multi-organ complications due to the presence of different comorbidities. Thus, multiple factors are involved that lead to the increased severity of COVID-19 risk factor in advanced age cancer patients [11]. 

The current study is one of the novel studies whose findings have suggested that cancer patients, particularly leukemia patients, may be more vulnerable to COVID-19 and have more pronounced lung structural damage due to the co-infection. Based on the results of this study related to leukemia patients who subsequently developed COVID-19, it is suggestive that these patients may be at a higher risk of long-term effects on the lungs. Other studies have shown increased COVID-19 mortality in hematologic malignancies; however, no studies, to the best of our knowledge, have looked at lung damage specifically as in this study. Studies have shown likely links between high rates of immune dysfunction in hematologic malignancies and increased mortality from COVID-19. Leukemia patients, in particular, are at a high risk for immune dysfunction, whether solely related to the primary disease process or therapy-related immunosuppression [31]. However, additional studies are required to understand the link between leukemia and COVID-19 as well as links between other types of cancers and COVID-19. 

However, there are some limitations of the current study including that the study was a non-randomized, retrospective analysis conducted on a small sample size. Further, some confounders such as tumor stage and COVID-19 infection severity were not included in the study, as they might affect the clinical outcomes of the cancer patients co-infected with COVID-19. Furthermore, due to retrospective study design, only fatal cases of COVID-19 cancer patients were reported in this study and a comparison between the COVID-19 infected cancer and non-cancer patients was not conducted. 

Thus, it is recommended to conduct future studies with a larger sample size and prospective study design to better explore the comparison between cancer and non-cancer patients infected with COVID-19 and their severe events. 

Further, it is also suggested that cancer patients should receive online medical counselling in addition to vigorous screening for COVID-19 to help identify critical cases for proper treatment. In cancer patients who do develop COVID-19, it may be necessary to avoid further immunosuppressive treatments or at least consider decreased dosages [8].

Thus, COVID-19 cancer patients reinforce several important considerations for clinical care, and thereby, there should be more emphasis on the urgent need for more research in this area.

## 5. Conclusions

This was a retrospective analysis study conducted on COVID-19 cancer patients to analyze the CT imaging changes before and after treatment and the extent of structural damage to the lungs post COVID-19 infection. It was found that, post COVID, there was a decrease of >50% in lung volume and a higher density in the form of HUs. Thus, it was concluded that COVID-19 infection had further detrimental effects on the lungs of cancer patients, thereby, decreasing their lung volume and increasing their lung density due to scar formation post COVID-19 infection. The study also supports the notion that patients with hematologic malignancies such as leukemia may be at particularly high risk for adverse outcomes from COVID-19 infection. Proactive strategies are required to reduce the likelihood of infection, and therefore, improved early identification of COVID-19 positivity in cancer patients is clearly warranted. Hence, increased follow-up is required to better understand the effects of COVID-19 on outcomes in cancer patients, considering the ability to follow specific cancer treatments. In general, during active treatment, cancer patients receive frequent follow-up which should be based on the extent of damage. However, we suggest that patients be followed weekly with labs and low-dose CT scans as needed until resolution of symptoms. If there is extensive fibrosis, monthly, then every 3 months, and then every 6 months followup is needed, but so far, no such protocol has been established. 

## Figures and Tables

**Figure 1 cancers-15-00651-f001:**
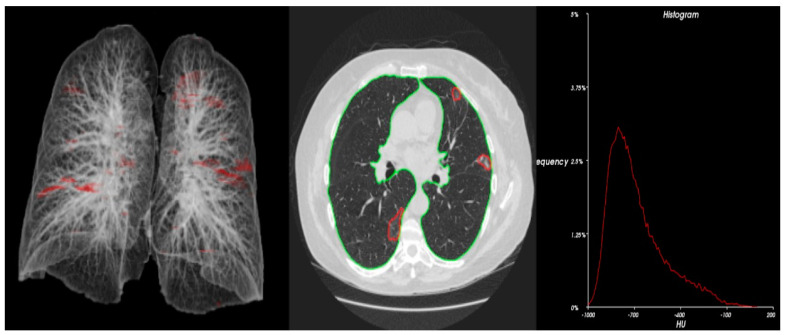
Radiometric 3D analysis (red circle part indicate COVID related damage in the lungs) and time density histograms constructed for Patient 2.

**Figure 2 cancers-15-00651-f002:**
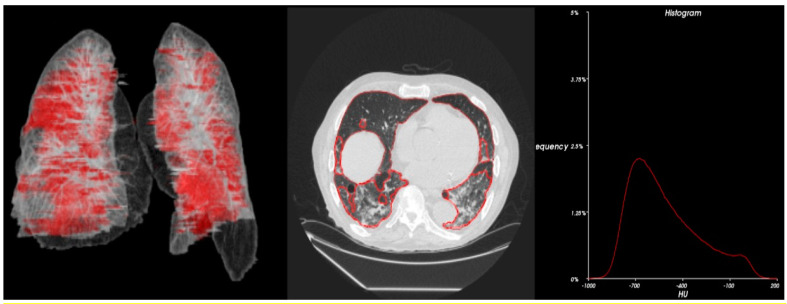
Radiometric 3D analysis (red circle part indicate COVID related damage in the lungs) and time density histograms constructed for Patient 3.

**Figure 3 cancers-15-00651-f003:**
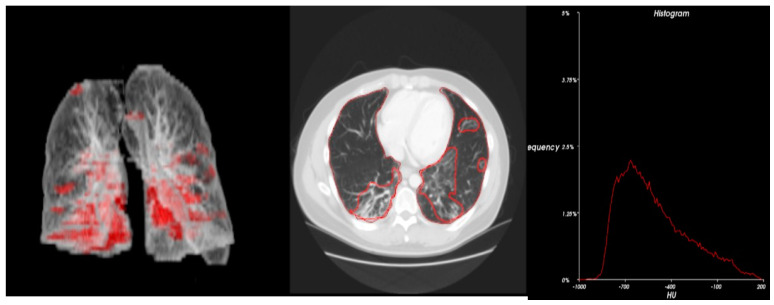
Radiometric 3D analysis (red circle part indicate COVID related damage in the lungs) and time density histograms constructed for Patient 6.

**Figure 4 cancers-15-00651-f004:**
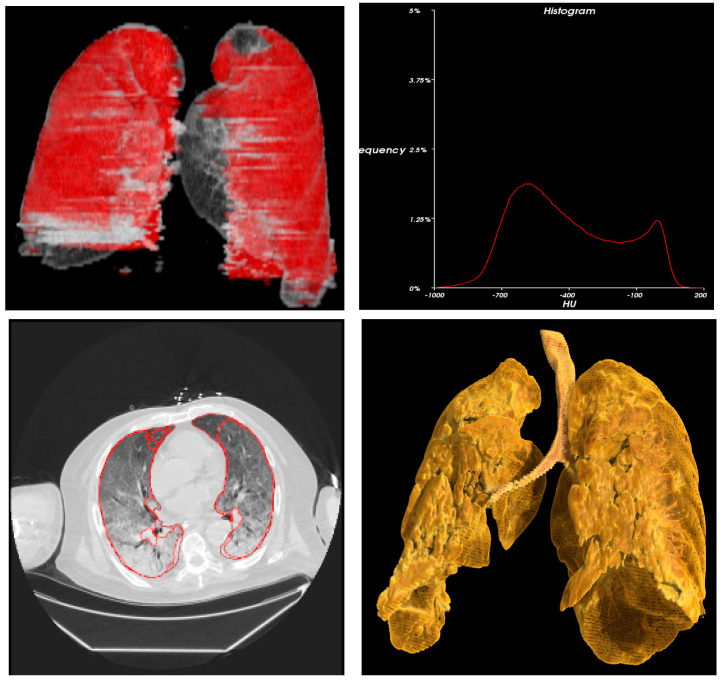
Radiometric 3D analysis (red circle part indicate COVID related damage in the lungs) and time density histograms constructed for Patient 7.

**Figure 5 cancers-15-00651-f005:**
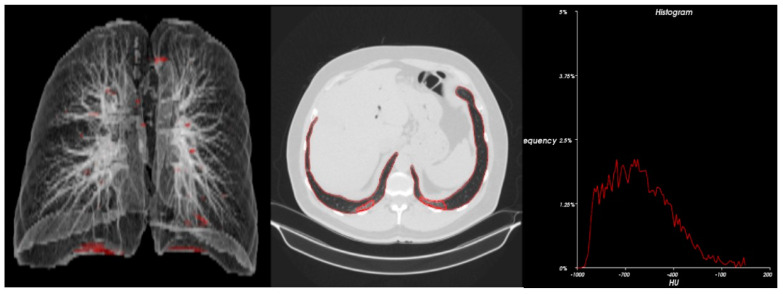
Radiometric 3D analysis(red circle part indicate COVID related damage in the lungs) and time density histograms constructed for Patient 15.

**Table 1 cancers-15-00651-t001:** Patients’ demographic characteristics (*n* = 28).

Demographic Characteristics		Frequency	Percentage
Age	Mean ± SD	55.6 ± 18.54	
Gender	Male	18	65.5
	Female	10	34.5

SD, standard deviation.

**Table 2 cancers-15-00651-t002:** Metastatic profile of cancer patients measured in Hounsfield units (HUs) and lung volume reduction (*n* = 28).

	Age (Years)	Gender	Lung (CC)	Lesion (CC)	Lesion (%)	<0.25 (HU)	>0.75 (HU)	Kurtosis
Patient 1	38	Male	3417.75	251.32	7.35	–668.40	−232.80	−1.06
Patient 2	64	Female	4842.12	28.34	0.59	−801.50	−577.70	0.89
Patient 3	79	Male	3983.27	1214.42	30.49	−677.90	−375.40	−0.23
Patient 4	54	Female	3822.75	63.32	1.66	−705.60	−288.80	−0.76
Patient 5	39	Female	4759.91	3030.76	63.67	−604.70	−198.80	−0.96
Patient 6	37	Male	3646.28	535.72	14.69	−679.20	−372.30	−0.18
Patient 7	73	Male	2998.78	2395.55	79.88	−590.30	−207.60	−1.00
Patient 8	59	Male	2772.49	239.01	8.62	−685.80	−241.20	−1.07
Patient 9	19	Male	4331.45	603.58	13.93	−628.40	−355.90	−0.59
Patient 10	34	Female	1625.70	725.48	44.63	−603.40	−209.60	−1.07
Patient 11	68	Female	4928.02	956.12	19.40	−760.10	−348.30	−0.66
Patient 12	66	Male	5173.26	10.36	0.20	−713.00	−447.40	−0.30
Patient 13	31	Female	2550.71	16.47	0.65	−763.63	−766.04	0.26
Patient 14	18	Male	3538.70	142.05	4.01	−704.75	−687.97	0.38
Patient 15	62	Male	2707.00	0.00	0.00	−710.00	−721.00	−1.06
Patient 16	51	Female	2387.56	313.39	13.13	−699.78	−722.27	0.75
Patient 17	83	Female	1984.52	1487.44	74.95	−413.59	−307.48	0.27
Patient 18	23	Male	1782.86	1031.47	57.85	−398.49	−450.33	1.24
Patient 19	83	Male	3749.23	73.12	1.95	−812.62	−810.27	0.34
Patient 20	65	Male	3474.40	238.29	6.86	−685.86	−693.05	0.51
Patient 21	62	Male	4233.50	146.46	3.46	−776.89	−784.54	0.36
Patient 22	73	Male	2699.41	185.86	6.89	−653.43	−689.60	0.27
Patient 23	64	Female	4507.96	51.79	1.15	−795.79	−813.74	0.27
Patient 24	34	Female	3341.41	0.00	0.00	−797.91	−810.44	0.26
Patient 25	74	Male	4202.10	230.08	5.48	−719.93	−700.45	0.45
Patient 26	63	Male	1842.53	70.84	3.84	−750.79	−787.42	0.24
Patient 27	63	Male	4486.89	351.70	7.84	−664.92	−691.91	0.57
Patient 28	70	Male	3090.97	1716.68	55.54	−493.92	−466.78	0.92

HU, Hounsfield unit.

**Table 3 cancers-15-00651-t003:** Correlation of metastatic profile post COVID infection (*n* = 15).

		Lung (CC)	Lesion (CC)	Lesion (%)
Patient 2	Pre	4842.12	28.34	0.59
	Post	4609.06	164.81	3.58
Patient 3	Pre	3983.27	1214.42	30.49
	Post	5821.05	457.45	7.86
Patient 6	Pre	3646.28	535.72	14.69
	Post	4399.97	242.51	5.51
Patient 7	Pre	2998.78	2395.55	79.88
	Post	3390.13	2970.58	87.62
Patient 15	Pre	2707.00	0.00	0.00
	Post	1455.00	0.00	0.00
Patient 16	Pre	2387.56	313.39	13.13
	Post	2598.47	243.58	9.37
Patient 18	Pre	1782.86	1031.47	57.85
	Post	2733.87	0.00	0.00
Patient 19	Pre	3749.23	73.12	1.95
	Post	3578.28	524.87	14.67
Patient 20	Pre	3474.40	238.29	6.86
	Post	2942.11	133.54	4.54
Patient 22	Pre	2699.41	185.86	6.89
	Post	3964.46	48.37	1.22
Patient 23	Pre	4507.96	51.79	1.15
	Post	4634.40	247.88	5.35
Patient 24	Pre	3341.41	0.00	0.00
	Post	1749.26	856.68	48.97
Patient 26	Pre	1842.53	70.84	3.84
	Post	3025.89	1271.02	42.00
Patient 27	Pre	4486.89	351.70	7.84
	Post	5492.09	7.94	0.14
Patient 28	Pre	3090.97	1716.68	55.54
	Post	4482.71	18.68	0.42

**Table 4 cancers-15-00651-t004:** Statistical analysis of pre and post COVID infection (*n* = 15).

*n* = 15	ΔLung Vol % (Post-Pre)/Pre	ΔLesion Vol (cc) (Post-Pre)/Pre	ΔLesion % (Post-Pre)
Mean	13.65%	−67.95	−3.92
Standard Error	34.23%	729.77	28.18

## Data Availability

The data presented in this study are available on request from the corresponding author.

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
