# Peer review of "COVID and Cancer: A Complete 3D Advanced Radiological CT-Based Analysis to Predict the Outcome"

_cancers, 2023, doi:10.3390/cancers15030651_

Round 1

Reviewer 1 Report

The study entitled as "COVID and Cancer: A complete 3D advanced Radiological CT-based analysis to predict the outcome" is of great significance as the prognosis of the disease in cancer patients is significantly different from the normal individuals infected with SARS-CoV-2. 

As the sample size is really small, and the subjects are not selected on the basis of their stage of the cancer. Additionally, The age parameter is also very much mixed. Hence, its difficult to draw the conclusion statistically as the age and gender also plays an important part in the prognosis of the COVID-19 disease. However, the article can be improved to reach at some level of results and discussion and I do have some comments and suggestions to get addressed before the publication of the article.

In the abstract:

Results: A total of 22,  cancer patients were diagnosed with COVID-19 infection.

In the background, Line no. 15, this information has been already given.

So, to avoid redundancy, this can be removed from the results. 

Line no. 22: Is it possible to provide the stage of the cancer. 

Line no. 26 to 28: the sentence structure is too complex, make it easy either by fragmentation 

or, remove the post COVID-19 infection at Line no. 28. 

Line no. 32: (SARS-CoV2) change to SARS-CoV-2 and change throughout the manuscript. 

Line no. 63 to 65: The information is not clear. Check what authors actually wants to convey to the readers. 

In the introduction section, authors are emphasising more on the differential prognosis between cancer patients or normal patients infected with SARS-CoV-2, 

Except from line no. 75 to 79, I did not see information related to the present study. Readers would like to see more information regarding the research conducted specifically why this research was conducted. 

Line no 82 to 83: 19 males, 10 females, the total will be 29, 

Please justify why it is saying 28 at the first. 

Table No.4:  Statistical analysis of pre and post-COVID infection (n=15), is not clear to me. Please check. 

Line no. 187-189: This has been mentioned earlier. 

Moreover, the study is of very small sample size and it is difficult to show statistical differences among the various parameters. I think all these must be present as a different section under the Limitations: 

Line no. 244: Hence, increased follow-up is required

What types of steps can be acquired to follow up the patients?

I suggest to describe some feasible steps and procedures here. 

At the end, I found at several points, the sentences are quite wordy, which can be simplified to increase the readability. 

Best wishes. 

Author Response

Thanks

Reviewer 2 Report

The paper entitled "COVID and Cancer: A complete 3D advanced Radiological CT-based analysis to predict the outcome" by Syed Rahmanuddin et al is enteresting article, well written. In my opinion for pubblication the serve shorten discussion section.

Author Response

Thanks

Round 2

Reviewer 1 Report

The authors have revised the manuscript sufficiently. It can be accepted for the publication.